# Synopsis of the Genus *Trichorondonia* Breuning, 1965 with Description of a New Species from China (Coleoptera: Cerambycidae) [note 1]

**DOI:** 10.3390/insects16070743

**Published:** 2025-07-21

**Authors:** Ruigang Yang, Jianhua Huang, Guanglin Xie

**Affiliations:** 1Guangxi Research Academy of Environmental Science, Nanning 530022, China; yangruigang276@163.com; 2Key Laboratory of Forest Bio-Resources and Integrated Pest Management for Higher Education in Hunan Province, Central South University of Forestry and Technology, Changsha 410004, China; caniscn@aliyun.com; 3Institute of Entomology, College of Agriculture, Yangtze University, Jingzhou 434025, China

**Keywords:** Acanthocinini, longhorn beetle, new species, taxonomic key, taxonomy

## Abstract

The genus *Trichorondonia* Breuning, 1965 currently comprises three species. The type species, *Trichorondonia hybolasioides* Breuning, 1965, is distributed in Laos and China. The other two species, *Trichorondonia pilosipes* (Pic, 1907) and *Trichorondonia kabateki* Viktora, 2024, are both endemic to China. This paper provides a brief revision of the genus and describes a new species: *Trichorondonia wenkaii* sp. nov. Additionally, the male of *Trichorondonia kabateki* is described for the first time with the new locality record, and a key to the four species is provided.

## 1. Introduction

*Trichorondonia* Breuning, 1965 belongs to the tribe Acanthocinini of the subfamily Lamiinae in Cerambycidae, with three known species to date. It was first established for a Laotian species, *Trichorondonia hybolasioides* Breuning, 1965 [1]. This genus is similar to *Neacanista* Gressitt, 1940, *Trichohoplorana* Breuning, 1961, and *Laoneacanista* Gouverneur & Chemin, 2023, but differs from them in having a body entirely covered with long erect hairs and/or bristles, a pronotum with a pair of rather small discal tubercles, and no clearly visible punctures. The genus also somewhat resembles the genus *Trichacanthocinus* Breuning, 1963, but can be distinguished from the latter in antennomere 3 being equal to antennomere 4 and being slightly shorter than the scape in length and the pronotum bearing small and pointed lateral spines and lacking discal tubercles. Xu et al. proposed the subgenus *Pogonocherus* (*Neopogonocherus*) Lazarev, 2021 as a junior synonym of *Trichorondonia* and transferred *Pogonocherus pilosipes* Pic, 1907 to *Trichorondonia* [2]. Viktora described the third species, *Trichorondonia kabateki* Viktora, 2024 in Sichuan, China [3].

In the present work, a new species, *Trichorondonia wenkaii* sp. nov., from Guangxi Zhuang Autonomous Region, China, is described and illustrated. A key to the known species of the genus is given.

## 2. Materials and Methods

All examined specimens are deposited in the following institutional or private collections:

**BPBM,** Bernice Pauahi Bishop Museum, Honolulu, HI, USA;

**CPV,** Collection of Petr Viktora, Kutná Hora, Czech Republic;

**MNHN,** Muséum national d’Histoire naturelle, Paris, France;

**ICYZU,** Insect Collection, College of Agriculture, Yangtze University, Jingzhou, Hubei, China.

The photographs of the habitus of *Trichorondonia wenkaii* sp. nov. were taken using a Canon 7D Mark II digital camera equipped with a Canon EF 100 mm f/2.8L IS USM lens (Canon (China), Beijing, China). The photograph of the holotype of *T. hybolasioides* was offered by Guiqiang Huang (Liupanshui Normal University, Guizhou, China), the photograph of the holotype of *Pogonocherus pilosipes* was offered by Maxim A. Lazarev (Tverskaya, Russia) and the photograph of the holotype of *T. kabateki* is copied from Petr Viktora (Kutná Hora, Czech Republic) with his permission. All photographs were processed and edited using Adobe Photoshop 2020 release.

Verbatim quotation is used here for all labels of the studied type specimens, and the label text is given in single quotation mark. Individual labels are separated by a semicolon, and data on different rows are separated by a single slash. Additional and explanatory comment by the authors is given in square brackets. Abbreviations are used in the text for label text: h for handwritten, p for printed.

This article is registered in ZooBank under the link http://zoobank.org/urn:lsid:zoobank.org:pub:28E18AAE-AAB8-4C6E-B230-DD044CAFA58D.

## 3. Results

***Trichorondonia*** **Breuning, 1965**

Chinese common name: 毛郎氏天牛属

*Trichorondonia* Breuning, 1965: 60 [1]; Rondon and Breuning 1970: 508 (catalogue) [4]; Breuning 1977: 112 (key) [5]; Breuning 1978: 63 (redescription) [6]; Hubweber et al. 2010: 213 (catalogue) [7]; Lin and Tavakilian 2019: 224 (catalogue) [8]; Danilevsky 2020: 298 (catalogue) [9]; Xu et al. 2021: 593 [2]; Viktora 2024: 431, 432 [3]. Type species: *Trichorondonia hybolasioides* Breuning, 1965, by original designation and monotypy.

*Pogonocherus* (*Neopogonocherus*) Lazarev, 2021: 63 [10]. Type species: *Pogonocherus pilosipes* Pic, 1907, by original designation and monotypy.

**Generic diagnosis.** Body small-sized, covered with greyish-white erect hairs and blackish-brown bristles. Antennae slender, longer than body; scape slightly fusiform, antennomere 3 about as long as scape, shorter than antennomere 4, antennomere 4 distinctly longer than subsequent segments. Eyes deeply emarginated, coarsely faceted. Frons distinctly transverse. Pronotum wider than long; lateral spines short and stout, curved backward; disc with a pair of small tubercles. Elytra convex, truncate or emarginate, each elytron with two median longitudinal carinae and a developed lateral carina behind humerus. Pro- and mesocoxal cavities closed. Legs moderately long, with femora distinctly swollen apically, meso- and metafemora distinctly clavate, claws divaricate.

**Distribution.** China, Laos.

***Trichorondonia pilosipes*** **(Pic, 1907)**

Figure 1a,i

Chinese common name: 毛足毛郎氏天牛

*Pogonochaerus* [*sic*] *pillosipes* [*sic*] Pic, 1907: 21 [11]. Type locality: “Chine Orientale”.

*Pogonocherus* (s. str.) *pilosipes* Pic: Aurivillius 1923: 332 (catalogue) [12]; Winkler 1929: 1210 (catalogue) [13]; Hua 1982: 112 (checklist) [14]; Lin and Tavakilian 2019: 361 (catalogue) [8]; Danilevsky 2020: 448 (catalogue) [9].

*Pogonocherus* (s. str.) *pillosipes* [*sic*] Pic: Plavilstshikov 1926: 155, 161 [15].

*Ponogocherus* [*sic*] (s. str.) *pillosipes* [*sic*] Pic: Gressitt 1951: 516 (fauna) [16].

*Pogonocherus* (*Eupogonocherus*) *pilosipes* Pic: Breuning 1963: 519 (catalogue) [17]; Breuning 1975: 28 (redescription) [18].

*Pogonocherus pilosipes* Pic: Hua 2002: 225 (catalogue, attribute to “Pic, 1923”) [19]; Hua et al. 2009: 465 (list, attribute to “Pic, 1923”) [20]; Hubweber et al. 2010: 31 (catalogue) [7].

*Pogonocherus* (*Neopogonocherus*) *pilosipes* Pic: Lazarev 2021: 63 (redescription) [10].

*Trichorondonia pilosipes* (Pic): Xu et al. 2021: 595 (new combination) [2]; Viktora 2024: 431 (distribution) [3].

**Type material examined.** Holotype male (MNHN), Chine Orientale. Examined from a photograph offered by Lazarev (sent to him by Gérard Tavakilian) (Figure 1a).

**Distribution.** China (Orient).

**Remarks.** Lazarev (2021) established the subgenus *Pogonocherus* (*Neopogonocherus*) based on *Pogonocherus pilosipes* Pic, 1907, and provided a redescription of this species. Subsequently, Xu et al. treated *Pogonocherus* (*Neopogonocherus*) as a junior synonym of *Trichorondonia* and transferred *P. pilosipes* to *Trichorondonia* [2].

***Trichorondonia hybolasioides*** **Breuning, 1965**

Figure 1b,j

Chinese common name: 毛郎氏天牛

*Trichorondonia hybolasioides* Breuning, 1965: 61 [1]; Rondon and Breuning 1970: 509 (catalogue) [4]; Breuning 1978: 63 (redescription) [6]; Hua 1987: 95 (distribution) [21]; Hua 2002: 235 (catalogue) [19]; Hubweber et al. 2010: 213 (catalogue) [7]; Lin and Tavakilian 2019: 224 (catalogue) [8]; Xu et al. 2021: 595 (remark) [2]; Viktora 2024: 431 (distribution) [3]. Type locality: Phontiou, Khammouan Province, Laos.

**Type material examined.** Holotype male (BPBM, BPBMENT0000008698): ‘*Trichorondonia* [p, black] n. g. [p, red]/hybolasioides [p, black]/Coll. J. A. Rondon Laos [p, yellowish-brown]/n. sp. Breuning [p, red] [label rectangular, yellowish-brown framed]; Phontiou/27-11-63 [h, blue, label rectangular, black-framed]; *Trichorondonia*/*hybolasioides*/mihi typ [h, blue]/Breuning dét. [p, black] [label rectangular, white]; 1 ex. 6. December 2013/J. Yamasako [h, black; label rectangular, white]’. Examined from a photograph offered by Guiqiang Huang (sent to him by James H. Boone) (Figure 1b).

**Distribution.** China (Guangxi); Laos (Khammouan).

**Remarks.** Xu et al. corrected the type locality mentioned in the original description, revising it from ‘région de Thakhek’ to ‘Phontiou’ [2]. This species closely resembles *T. pilosipes* (Pic, 1907) but differs in having distinctly acuate elytral outer apical angles (Figure 1j), unlike the obtusely dentiform projections seen in the latter (Figure 1i).

***Trichorondonia kabateki*** **Viktora, 2024**

Figure 1c–f,k

Chinese common name: 卡巴毛郎氏天牛

*Trichorondonia kabateki* Viktora, 2024: 432 [3]. Type locality: Jiuzhaigou, Sichuan, China.

**Description.** Male, body length 8.5–8.8 mm, humeral width 3.0–3.1 mm. Body chestnut to blackish-brown, covered with greyish-yellow, greyish-white, and blackish-brown pubescence, along with sparse blackish-brown and greyish-white bristles, legs and ventral side covered with long greyish-white hairs. Head blackish-brown, with the frons and areas behind the eyes adorned with long greyish-yellow bristles, while vertex covered with long blackish-brown bristles. Antennae mostly reddish-brown, with apex of each antennomere slightly darker; scape covered with sparse long greyish-yellow bristles and dotted with sparse coarse punctures; antennomeres 1–10 fringed with long greyish-yellow bristles below, antennomeres 3–11 covered with greyish-white pubescence basally. Pronotum mostly dark reddish-brown, densely covered with greyish-yellow pubescence; disc clothed with long sparse greyish-yellow bristles, decorated with a curved blackish-brown stripe on each side of midline. Elytra mostly dark reddish-brown, patterned with greyish-yellow, greyish-white and blackish-brown pubescence; elytral base mostly dark, forming a blackish-brown patch in an approximately inverted triangular shape, followed by a “V”-shaped band composed of greyish-yellow and greyish-white pubescence; each elytron provided with greyish-white oblique pubescent spots at approximately apical fifth and a central tubercles covered by a cluster of long black setae at base; elytral surface dotted with greyish spots interspersed with blackish spots along suture and longitudinal carinae. Legs dark reddish-brown, covered with greyish-yellow pubescence, tibiae blackish-brown apically.

Head short, with indistinct punctures; frons transverse, with a smooth and extremely fine median longitudinal sulcus; vertex slightly depressed between eyes, provided with a pair of small and inconspicuous tubercles covered with hairs behind eyes; lower eye lobe about equal to gena in length. Antennae distinctly longer than body, with terminal three segments surpassing elytral apex; scape sparsely punctate. Pronotum slightly wider than long, with lateral tubercles short and blunt; disc convex, with a small bump on each side of midline. Elytra about 2.0 times as long as humeral width, with sides subparallel, gradually converging apically; disc with coarse and sparse punctures at base, and unobvious at apical fifth; each elytron provided with a distinct bristly tubercle at centre of base, a developed lateral carina after humerus extending to subapex, accompanied mesially by two longitudinal carinae which do not attain margins; apex nearly transversely truncate, with outer angle and internal angle not projecting. Legs moderately long, with femora club-shaped and claws divaricate.

**Figure 1 insects-16-00743-f001:**
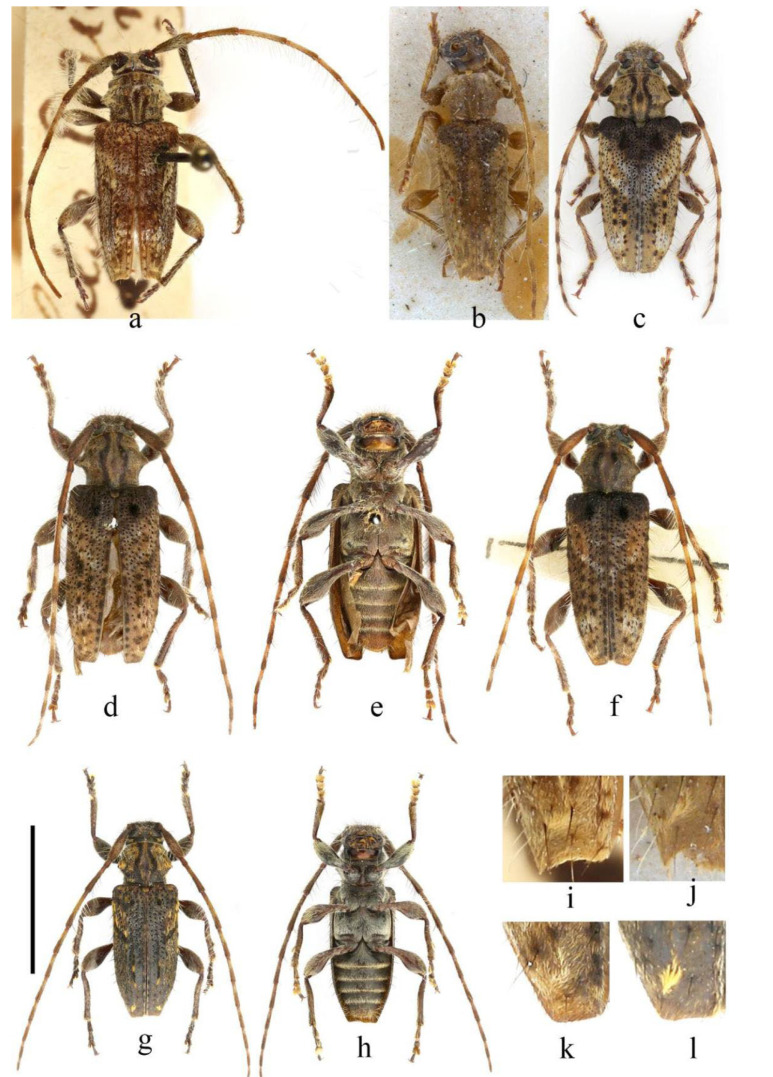
Habitus of *Trichorondonia* spp. (**a**,**i**) holotype male of *T. pilosipes* (Pic, 1907). (**b**,**j**) Holotype male of *T. hybolasioides* Breuning, 1965. (**c**–**f**,**k**) *T. kabateki* Viktora, 2024: (**c**) Holotype female; (**d**,**e**) male from Fang County; (**f**,**k**) male from Yichang. (**g**,**h**,**l**) Holotype male of *T. wenkaii* sp. nov. (**i**–**l**) elytral apex. Scale bars for (**d**–**h**): 5 mm.

**Type material examined.** Holotype female (CPV): ‘*Trichorondonia kabateki* sp. nov./HOLOTYPUS/P. Viktora det., 2024 [p, label red]; C CHINA, Sichuan prov./31 km NW Jiuzhaigou, 2048 m/Zhongcha vill. 33°18.615′ N,/103°58.537′ E, 3. and 5. VII. 2012/P. Kabátek lgt.’ (Figure 1c).

**Other material examined.** China: 1 ♂, Hubei Province, Fang County, Qiaoshang Township, Xiping Village, alt. 1070 m, 20 June 2008, Guang-Lin Xie leg.; 1 ♂, Hubei Province, Yichang city, Dalaoling National Nature Reserve, alt. 1200–1300 m, 30 April 2008, Guang-Lin Xie leg.

**Distribution.** China: Sichuan, Hubei (new provincial record).

**Remarks.** This species differs from *T. pilosipes* and *T. hybolasioides* in possessing a pair of distinct basal elytral tubercles covered by a cluster of long black setae and rounded (non-protruding) elytral outer apical angles (Figure 1k).

***Trichorondonia wenkaii*** **sp. nov.**


http://zoobank.org/urn:lsid:zoobank.org:act:2C5F3DF1-BF5C-4447-9A9D-7028D8A3EFEB


Figure 1g–l, Figure 2

Chinese common name: 文凯毛郞氏天牛

**Description.** Male. Body length 6.7 mm, humeral width 2.2 mm. Body mostly blackish-brown, clothed with greyish-yellow and black pubescence, as well as greyish-white and black setae forming distinct patterns. Head blackish-brown, clothed with blackish pubescence and erect setae on frons, greyish-yellow pubescence and greyish-white erect setae on gena and blackish-brown pubescence on vertex behind eyes. Antennae dark reddish-brown; antennomeres 1–7 ventrally fringed with sparse greyish-white hairs; scape clothed with sparse black erect setae on dorsum; antennomeres 3–11 annulated with greyish-yellow pubescence near base. Pronotum blackish-brown, clothed with dense greyish-yellow pubescence on both sides of anterior half and rather sparse on both sides of posterior half; each side of midline adorned with a conspicuous curved longitudinal strip of blackish-brown pubescence, internally bordered by greyish-yellow pubescent margins and exhibiting an outward curvature medially. Elytra nonuniformly covered with greyish-yellow pubescence, denser at basal half, behind each humerus, with a distinct broad oblique pubescent patch extending to just before middle, immediately followed by a narrow blackish-brown longitudinal stripe; each elytron provided with a cluster of black setae presents at central base, several alternating yellow and black pubescent spots on two longitudinal carinae in middle section of disc, a prominent greyish-yellow pubescent spot before apex, as well as several vague spots alternating in greyish-yellow and blackish-brown at about apical third of suture; whole surface sparsely clothed with black setae, while white along lateral edges. Ventral surface clothed with greyish-white pubescence, denser on metaventrite, and forming a distinct greyish-white pubescent transverse band at each apical margin of abdominal ventrites. Legs dark reddish-brown, clothed with greyish-yellow pubescence and greyish-white erect setae; femora clothed with rather dense greyish-yellow pubescence on dorsal surface.

Frons transverse, lower eye lobes about as long as gena, vertex slightly depressed between eyes and with a pair of rather small hair-covered tubercles behind eyes. Antennae distinctly longer than body; antennal tubercle widely separated; scape slender and slightly fusiform, slightly longer than antennomere 3, with sparse, coarse and shallow punctures; antennomere 4 longest, antennomeres 5–11 gradually decreasing in length. Pronotum slightly wider than long, lateral spine short, conical, slightly directed backwards; disc convex, with a small tubercle on each side of midline and a shallow median sulcus posteriorly. Elytra about 2.0 times as long as humeral width, with sides subparallel and apical quarter slightly narrowed; apices transversely truncate, with rounded sutural angles and lateral angles that are not prominent; surface coarsely punctate, arranged somewhat regularly in rows, sparser at base, indistinct at apical sixth of sides and apical third of middle; each elytron provided with four distinct longitudinal carinae: two behind humerus, with outer one extending to about apical seventh and inner one to about apical two thirds; and other two at centre, with the outer one being longer. Legs moderately long, with femora slender in basal half and distinctly enlarged in apical half, and tibiae slightly curved basally.

**Male genitalia.** Tergite VIII longer than wide, about 1.7 times as long as wide at apex, subrectangle, nearly straight apically, with apical third of sides and both sides of apex slightly darker in colour and sparsely covered with long black setae (Figure 2a,b). Spiculum relictum shorter than a half of spiculum gastrale, spiculum gastrale shorter than ringed part of tegmen. Tegmen slightly bent in lateral view, parameres elongate, each about 4.5 times as long as width, apex with sparse setae. Median lobe moderately curved in lateral view, about as long as tegmen; median struts about half length of median lobe (Figure 2c–e). Apex of ventral plate rounded; endophallus long, mostly membranous.

**Female.** Unknown.

**Material examined.** Holotype: China: ♂(ICYZU), Guangxi Zhuang Autonomous Region, Leye county, Xinhua town, Dianping village, 24.6806°N, 106.7649°E, alt. 769 m, 23 April 2024, Gunag-Lin Xie leg.

**Differential diagnosis.** This new species differs from *T. pilosipes* and *T. hybolasioides* in having elytra with rounded lateral apical angles (Figure 1l) and a vertex with blackish-brown pubescence medially behind the eyes, whereas the latter two exhibit distinctly protruding lateral apical angles in the elytra (Figure 1i,j) and greyish-yellow pubescence on the vertex behind the eyes.

It also can be easily distinguished from *T. kabateki* by the antennae being ventrally fringed with sparse hairs only on segments 1–8, the greyish-yellow pubescence on the pronotum being unevenly distributed and particularly sparse in the posterior half of the elytra, with rather thin greyish-yellow pubescence, hardly visible greyish-white pubescence, elongated blackish-brown spots on the elytral longitudinal carinae, and a small tuft of black setae at the centre of the elytral base where there is no obvious tubercle. In contrast, *T. kabateki* has the antennae ventrally fringed with hairs on antennomeres 1–10, the pronotum and elytra clothed with dense greyish-yellow pubescence, and the base of the elytra provided with a prominent inverted triangular dark brown area and a pair of tubercles with dense black bristles.

**Etymology.** The new species is named after Professor Wang Wenkai, a cerambycid taxonomist and the mentor of the corresponding author.

**Figure 2 insects-16-00743-f002:**
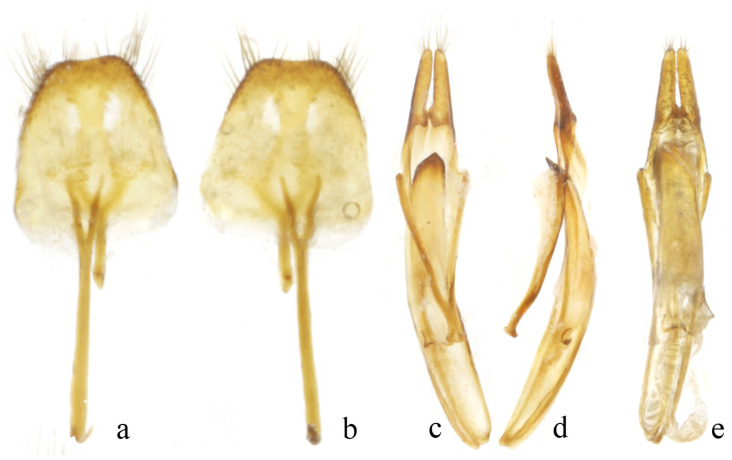
Male genitalia of *Trichorondonia wenkaii* **sp. nov.** (**a**,**b**) Tergite VIII and sternite VIII. (**c**–**e**) Tegmen + median struts. (**a**,**c**) Ventral view. (**b**,**e**) Dorsal view. (**d**) Lateral view.

Key to the species of *Trichorondonia* Breuning

1.Apices of elytra emarginate..................................................................................................2−Apices of elytra evenly rounded..........................................................................................32.Outer angles of elytral apices slightly prominent and obtuse........................***T. pilosipes***−Outer angles of elytral apices strongly prominent, acute.......................***T. hybolasioides***3.Underside of 1–10 antennomeres fringed with setae; sides of pronotum evenly covered with greyish-yellow pubescence; elytra pubescent with relatively dense greyish-yellow hairs intermixed with rather conspicuous greyish-white hairs, forming a distinct inverted triangular black macula at base..................................................***T. kabateki***−Underside of 1–8 antennomeres fringed with setae; pronotum clothed with unevenly distributed greyish-yellow pubescence at sides, particularly sparse basally; elytra irregularly covered with relatively thin greyish-yellow pubescence, greyish-white pubescence inconspicuous and not composing a distinct inverted triangular macula at base...........................................................................................................***T. wenkaii* sp. nov.**

## Data Availability

The original contributions presented in this study are included in the article. Further inquiries can be directed to the corresponding author.

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
