# Peer review of "Synopsis of the Genus Trichorondonia Breuning, 1965 with Description of a New Species from China (Coleoptera: Cerambycidae)â€"

_insects, 2025, doi:10.3390/insects16070743_

Round 1
Reviewer 1 Report
Comments and Suggestions for Authors
The study seems to be publishable in this form with minor revision. Although Trichorondonia Breuning, 1965 includes only 4 species with one new species described in this study. However, as the authors stated in their study, it is quite clear that these 4 species can belong to 2 subgenera, especially based on the structure of the elytral apex and other elytral features. In this case, it would be appropriate for the authors to define a new subgenus for the species T. kabateki and T. wenkaii. The decision on whether to make the definition in this study should still be left to the authors.
The necessary corrections are presented in the attached file.

Author Response
comment 1: In this case, it would be appropriate for the authors to define a new subgenus for the species T. kabateki and T. wenkaii. The decision on whether to make the definition in this study should still be left to the authors.
Response 1: the subgenus usually are proposed for the species in large genus. Since the genus Trichorondonia Breuning, 1965 has only four species including the new species described in this manuscript, we think there is no need to establish a new subgenus to contain T. kabateki and T. wenkaii.
Comment 2: The necessary corrections are presented in the attached file.
Response 2: We have accepted all suggestions by the reviewer 1.

Reviewer 2 Report
Comments and Suggestions for Authors
I think it is a well written taxonomic MS. I agreed that the new species is new. Based on what are available, it is reasonable to keep the version. The following is only discussion, not requirement. Years ago, Mr. Wen-Xuan Bi had told me that Trichorondonia hybolasioides Breuning, 1965 should be a synonym of Trichorondonia pilosipes (Pic, 1907). He has material from China on this group. However, I did not have any material so I could not test his conclusion. If you want, you might try to contact with him for cooperation.
one tiny form problem, on page 3, under the citation of pilosipes, one pilosipes should not be bold and should be italic.
Author Response
Comments 1: ......Years ago, Mr. Wen-Xuan Bi had told me that Trichorondonia hybolasioides Breuning, 1965 should be a synonym of Trichorondonia pilosipes (Pic, 1907). ...... If you want, you might try to contact with him for cooperation.
Response 1: At the moment, we do not be able to test this hypothesis because no material is available for us to examine and compare.
Comments 2: one tiny form problem, on page 3, under the citation of pilosipes, one pilosipes should not be bold and should be italic.
Response 2: We have corrected this error according to the suggestion of the reviewer 1 and 2.

Reviewer 3 Report
Comments and Suggestions for Authors
Dear authors,
This manuscript provides a brief review of the genus Trichorondonia and describes the new species Trichorondonia wenkaii sp. nov. It is a traditional taxonomic study, produced in accordance with the conventions of such studies and largely in line with the requirements of the Code of Zoological Nomenclature. However, the following remarks should be taken into account and the text corrected accordingly:
- The abstract (line 22) should outline the characters that distinguish Trichorondonia wenkaii sp. nov. from other similar species, the location where the species was found, and where the holotype will be stored.
- As taxonomic acts published in electronic journals must be registered at http://zoobank.org before publication, and as the reference must be included in the article, it is necessary to insert these references into the manuscript. The first reference, regarding the registration of the article in Zoobank, should be inserted at the end of the 'Materials and Methods' section (line 68), as follows:
This article is registeresd in ZooBank under the link http://zoobank.org/urn:lsid:zoobank.org:pub:28E18AAE-AAB8-4C6E-B230-DD044CAFA58D.
A second reference relating to the description of the new taxon is inserted immediately below its name when it is being described (line 202), in the following form:
Trichorondonia wenkaii sp. nov.
http://zoobank.org/urn:lsid:zoobank.org:act:2C5F3DF1-BF5C-4447-9A9D-7028D8A3EFEB
Figure 1g, h, l
- In the Key to the species of Trichorondonia Breuning, some terms are labelled with words that are not used when describing insects (see edit in the manuscript file). In general, the key can be rewritten as follows:
1 Apices of elytra emarginate .....................................................................2
- Apices of elytra evenly rounded..............................3
2 Outer angles of elytral apices slightly prominent, obtuse ...........................................................T. pilosipes
- Outer angles of elytral apices strongly prominent, acute ..........................................................T. hybolasioides
3 Underside of 1–10 antennomeres fringed with setae ; sides of pronotum evenly covered with greyish-yellow pubescence; elytra pubescent with relatively dense greyish-yellow hairs intermixed with rather conspicuous greyish-white hairs, forming a distinct inverted triangular black macula at base (please, check colours there).....................................................................................................................T. kabateki
- Underside of 1–8 antennomeres fringed with setae; pronotum clothed with unevenly distributed greyish-yellow pubescence at sides, particularly sparse basally; elytra irregularly covered with relatively thin greyish-yellow pubescence, greyish-white pubescence inconspicuous and not composing a distinct inverted triangular macula at base..........................................T. wenkaii sp. nov.

Please check the entire text for grammatical errors.
Author Response
Comments 1: The abstract (line 22) should outline the characters that distinguish Trichorondonia wenkaii sp. nov. from other similar species, the location where the species was found, and where the holotype will be stored
Response 1: We have outline the content the reviewer suggested.
Comments 2: As taxonomic acts published in electronic journals must be registered at http://zoobank.org before publication, and as the reference must be included in the article, it is necessary to insert these references into the manuscript.
Response 2: We have inserted the references the reviewer suggested into the corresponding place in the manuscript.
Comments 3: In general, the key can be rewritten as follows: ......
Response 3: We have rewritten the key according to the suggestion of the reviewer.
